# Assessment and Management of Maxillary Labial Frenum—A Scoping Review

**DOI:** 10.3390/diagnostics14161710

**Published:** 2024-08-06

**Authors:** Ryan Kinney, Richard C. Burris, Ryan Moffat, Konstantinia Almpani

**Affiliations:** College of Dental Medicine, Roseman University of Health Sciences, 10920 S River Front Pkwy, South Jordan, UT 84095, USA; rkinney797@student.roseman.edu (R.K.); rburris177@student.roseman.edu (R.C.B.); rmoffat@roseman.edu (R.M.)

**Keywords:** frenectomy, frenotomy, lip tie, maxillary frenum, upper lip frenum

## Abstract

Background: The maxillary labial frenum (MLF) is a soft tissue fold connecting the upper lip to the alveolar process. Abnormal attachment can cause periodontal, functional, and esthetic problems. Differential diagnosis is important and can prevent unnecessary interventions. This study aims to summarize the current evidence on the assessment and management of abnormal MLF. Methods: A thorough review of the literature was conducted. Five online databases were searched for relevant peer-reviewed human studies. Article screening and data extraction were performed independently by two reviewers using predefined inclusion/exclusion criteria. Information about article type, study design, participants’ characteristics, interventions, and outcomes was extracted and synthesized. Results: 52 articles met the review criteria. MLF is a dynamic structure characterized by a wide normal morphological variation. MLF assessment in infants has not been standardized. Studies in pre-adolescents reported a change in the thickness and position of the MLF observed over time, resulting in a lower prevalence of abnormal MLF morphology. Studies in adolescents and adults reported variable differential diagnosis criteria. Lasers appear as the most advantageous frenectomy modality. Conclusions: There is a need for more objective MLF diagnostic protocols and treatment guidelines, which could prevent unnecessary surgical interventions.

## 1. Introduction

The maxillary labial frenum (MLF) is a soft tissue fold connecting the upper lip to the alveolar process, consisting primarily of connective tissue and epithelium [1]. It originates histologically from the residual central cells of the vestibular lamina. It is a dynamic structure and is subjected to changes in shape, size, and position, staying in balance with the normal growth and development of the maxilla and the orodental function [2]. It provides support and stability to the lip and keeps the upper lip in harmony with the growing bones of the maxilla [3,4]. Hence, it plays an important role in the regulation of facial growth [2,3].

The maxillary labial frenum has fibrous tissue moving in an anteroposterior direction and merges with the submucosal fibers of the upper lip [3]. It also encompasses the septo-premaxillary ligament that serves as a means of transmitting the septal growth force to the premaxilla. It also encompasses striated fibers of the nasolabial muscles. In young children, the frenum is generally wide and thick, becoming thinner and smaller during growth, and eventually, it tends to diminish in size and importance [5]. The frenum tends to migrate apically due to primary incisor eruption, maxillary sinus development, and vertical growth of the alveolar process [6]. Therefore, inadequate muscular reconstruction and mutilation of the labial frenum could result in facial growth abnormalities.

On the other side, abnormal frenum attachment penetrating the papilla is considered as pathological because it is associated with midline diastema, gingival recession, loss of the papilla, interdental bone loss, poor lip mobility, difficulty in brushing, speech problems, malalignment of teeth, and closure of diastema during orthodontic treatment [3]. It has also been associated with increased dental plaque retention, which may lead to periodontal complications locally [4]. Finally, abnormal maxillary frenal attachments may act as a hindrance for the upper lip seal, thus making it difficult for the infants to breastfeed [4,7,8,9].

Due to the dynamic nature, diverse morphology, and attachment types, accurate differential diagnosis and management in different ages and dental developmental stages is critical and can dictate the need for or prevent unnecessary interventions. The aim of this review is to summarize the currently existing evidence on the differential diagnosis and management of abnormal MLF.

## 2. Materials and Methods

### 2.1. Protocol

The protocol was developed using the methodological framework for scoping reviews proposed by the Joanna Briggs Institute (https://jbi.global/scoping-review-network/resources, accessed on 5 July 2023) but has not been registered online. A systematic approach was followed for the conduction of this review following the Preferred Reporting Items for Systematic Reviews and Meta-analyses extension for scoping reviews (PRISMA–ScR). The specific type of review was selected because it allows for a systematic review of the literature with the aim to provide an overview of the available research evidence and elucidate potential gaps in the existing knowledge.

### 2.2. Eligibility Criteria 

Eligible articles included studies on healthy human participants of any age and sex/gender. Studies on subjects with craniofacial syndromes or cleft lip and/or palate were excluded because of other potential confounding factors that could interfere with the assessment and/or management of the MLF. No studies were excluded based on the geographic location of their conduction or details about the specific study setting. Only primary research studies were included. Table 1 provides a more detailed outline of the eligibility criteria for article selection.

### 2.3. Information Sources and Search 

To ensure a comprehensive search, the following databases were searched in August 2023: MEDLINE (PubMed), Web of Science Core Collection (Clarivate), Scopus (Elsevier), and Cochrane Library (Wiley) with the use of English Language and Humans filters. Grey literature and further non-database searching were done using Dissertations and Thesis Global (ProQuest), and a keyword search of Google Scholar that harvested the first 100 results. The complete search strategy for all databases and search engines is provided in Appendix A. Duplicates were removed using the duplicate removal tool in EndNote X7 citation manager software. Upon selection of the eligible studies, two authors (RK and RB) independently reviewed the references for each article included in the review as well as the references of relevant systematic reviews and/or meta-analyses to identify other potentially relevant studies for inclusion. The most recent search was executed on 4 August 2023.

An example search strategy that was used is the PubMed full electronic strategy:

(labial frenum*) OR (labial frena) OR (labial frenulum) OR (labial frenula) OR (upper lip frenum*) OR (upper lip frena) OR (upper lip frenulum*) OR (upper lip frenula) OR (upper frenum*) OR (upper frena) OR (upper frenulum)* OR (upper frenula) OR (maxillary frenum*) OR (maxillary frena) OR (maxillary frenulum*) OR (maxillary frenula) OR (maxillary labial frenum*) OR (maxillary labial frena) OR (MLF*)) AND ((diagnosis) OR (assessment*) OR (evaluation) OR (management) OR (frenotomy*) OR (frenectomy*) OR (referral*) OR (health impact)).

### 2.4. Selection of Sources of Evidence and Data Charting Process

Two authors (RK and RB) screened the titles and abstracts of the identified articles based on the predefined inclusion and exclusion criteria. If eligibility could not be decided by title or abstract, the full text of the article was retrieved. The decisions related to the initial screening of the articles were compared and any disagreements were resolved through discussion.

Data were extracted from the studies included in the scoping review by two independent reviewers (RK and RB) using a data extraction tool developed for this study. The extracted data included details about the participants, concept, context, study methods, and key findings relevant to the review questions. Data extraction was reviewed by a senior reviewer (KA).

### 2.5. Data Items and Synthesis of the Results

The data were summarized by two reviewers (RK and RB) and reviewed by the senior reviewer (KA). The characteristics of the included studies and the extracted information from each study are presented in tabular form as well as in the form of a narrative summary.

## 3. Results

### 3.1. Selection, Characteristics of Sources of Evidence, and Summary Results 

Of the 598 records identified by the initial search and the removal of 149 duplicates, 449 studies remained for screening. After reviewing their titles and abstracts, 54 studies qualified for retrieval and further assessment based on their full texts. Six of these studies had to be excluded based on information acquired from the full text. Four additional eligible studies were detected via a manual bibliographic reference search of the included studies. Fifty-two studies were eventually included in the review. Figure 1 outlines the study selection procedure.

Regarding the study types, there were 28 prospective cross-sectional studies, seven prospective randomized clinical trials, six prospective case-control studies, five retrospective cohort studies, four retrospective case-control studies, and two retrospective cross-sectional studies. Table 2 summarizes the main characteristics of the included studies, whereas Table 3 includes the objectives, participant characteristics, and main outcomes for each of the included studies.

### 3.2. Synthesis of the Results 

Publication dates ranged from 1977 to 2023. Most studies (39/53) evaluated participants with permanent dentition status, including adolescents and adults. In 24/53 studies, participants in a mixed dentition stage were part of the cohort. One study in toddlers in a primary dentition stage and eight studies in infants were also retrieved. Finally, there was one study with a mixed-age cohort with no further details on the dental developmental stage of the participants and one study with no pertinent participant characteristics information. The two main classification systems used in the included studies were Sewerin’s and Placek’s [24]. Sewerin’s MLF classification is based on the clinical morphology of the frenum: “Simple frenum”, “Persistent tectolabial frenum (PTF)”, “Simple frenum with an appendix”, “Simple frenum with a nodule”, “Double frenum”, “Frenum with a nichum”, and “Frenum with two or more variations at the same time”. Placek’s classification is based on the attachment site [5,24]. Specifically: Type I—mucosal frenal attachment, the frenal fibers are attached to the mucogingival junction; Type II—gingival frenal attachment, the fibers are inserted within the attached gingiva; Type III—papillary frenal attachment, the fibers extend into the interdental papilla; Type IV—papillary penetrating frenal attachment, the frenal fibers cross the alveolar process and extend to the palatine papilla. Types III and IV are more likely to be associated with complications and require intervention.

According to the results of the included studies, there is a diverse MLF morphology, which is largely associated with normal anatomic variations. The MLF morphology is subjected to variations in shape, size, and position during the different stages of growth and development. The simple type of frenum is the most common, whereas double freni, freni with nichum, and bifid freni are relatively rare. Bulky nodules or appendices on the freni and PTFs, though not considered pathological, may interfere with oral hygiene maintenance and thus adversely affect periodontal health. In such cases, the treatment options are removal of the nodule/appendix, or repositioning of the frenal insertion (in case of PTF), or even frenectomy. The prevalence of simple frenum increases from primary dentition to permanent dentition, whereas PTF decreases as age increases. A frenum’s nodule and appendix are considered as developmental remnants that show no pathological potential and do not need any investigation and treatment procedures. There was no significant association of frenal type with gender.

In a recent study by Kramer et al. [4] among preschoolers, morphological abnormalities were found in more than 1/5 of their cohort of over 1000 children, and ¼ of them exhibited an abnormal frenal attachment. However, both morphological abnormalities and abnormal attachment rates reduced significantly with the increase in age. In another study by Díaz-Pizán et al. [18] in a large cohort of 1,355 children in primary dentition, it was also noted that the level of gingival insertion moved apically with age and that the midline diastema was wider in younger children and significantly decreased with age.

Concerning the differential diagnosis in the infant stages, there seems to be a lack of consensus in the literature regarding whether MLF frenotomy improves breastfeeding outcomes. There is controversy on the identification, classification, and subsequent significance of the superior labial frenulum in newborns, and when the presence of a frenulum is “lip-tie”. The examination of the MLF is not part of the routine newborn clinical examination. Subsequently, the typical versus atypical appearance of this frenulum is not known. Nor is it known whether this frenulum has any functional consequences relating to its appearance or attachment. The primary justification for frenectomy procedures in infants is to facilitate and improve breastfeeding. However, there is little evidence that certain appearances of the labial frenula have any impact on latching or feeding.

In a study by Haischer-Rollo et al. [20], the MLF attachment site grade was evaluated utilizing a modified existing system, followed by an investigation of the correlation with breastfeeding outcomes. No association was detected between MLF attachment grade and objective breastfeeding-related outcomes. Their findings do not support labial frenectomy based on the clinical appearance of the MLF, and the authors highlight the need for a more robust functional grading system. In a study by Diercks et al. [19], a significant association was identified between maternal stress surrounding feeding, reduced breastfeeding self-efficacy, and frenotomy. According to the authors, this association raises important considerations regarding the impact of a frenectomy on maternal stress and perceptions, and how this may impact breastfeeding outcomes.

When MLF is associated with midline diastema, some of the articles suggested that the diastema should be initially closed with fixed orthodontic appliances, and an MLF frenectomy should be conducted as a second step. However, diastema closure after MLF with no orthodontic intervention was also investigated in a study by Baxter et al. [11], including participants in primary and mixed dentition stages with a diagnosis of hypertrophic MLF. According to their results, in 74.5% of patients in primary dentition and 75% of patients in mixed dentition, a preoperative diastema > 2 mm improved to <2 mm width post-operatively. Only some of the articles that dealt with surgery intervention included orthodontic intervention in their inclusion criteria. However, only 13 of these articles reported pre-set criteria for when to proceed with surgical intervention.

In some of the older studies in this review, MLF frenectomies were performed with the use of a scalpel-based procedure. For the conventional scalpel technique, hemostat pliers are used to grasp the frenum and then excise it with a surgical blade, leaving a wide wound area after the excision, which is closed with surgical sutures. Scalpel techniques have improved over the years to minimize scarring and post-operative discomfort. An example of these technical modifications is the paralleling technique. This technique involves two paralleling incisions made on the side of the ridge of the frenum, which reduces the removal of excess mucosal tissue. After that, a deep dissection of the muscle fibers is performed to remove the connective tissue attachment. This results in a relatively narrower wound area. Primary closure is also achieved with sutures. In addition, Z-plasty, which is a common transposition flap technique that was initially developed in plastic surgery in revision procedures to address scars, is a proposed scalpel technique for MLF frenectomies.

However, in more recent years, dental lasers have been thoroughly used as an alternative surgical modality. The main types of soft tissue lasers used for MLF frenectomies were CO_2_, Diode, Nd:YAG, Er:YAG, and Er,Cr:YSGG. Frenectomies performed by lasers utilize different surgical techniques compared to those of conventional scalpel methods. For example, horizontal or direction-based incisions in general are not seen with lasers, and because of the characteristics of lasers, specific incision types are not relevant. Dependence on the anatomy of the patient and where the frenum attachment is located are target points for frenectomy procedures utilizing lasers. Lasers utilize a different approach involving specific types of incisions based on frenum attachment and esthetic concerns. Comparing laser frenectomy to conventional techniques reveals significant differences in incision quality, tissue manipulation, and patient outcomes. Laser frenectomy offers superior precision, minimal tissue trauma, reduced bleeding, faster procedure times, and faster healing times compared to those of conventional methods. Additionally, the reduced risk of complications, such as bleeding, infection, and pain post-operatively makes laser frenectomy a preferred choice for many clinicians [10,12,13,22,25,32,34,35,40,48,49].

## 4. Discussion

### 4.1. Summary of Evidence

Based on a recent consensus report of the American Academy of Otolaryngology-Head and Neck Surgery, there was a consensus on the fact that upper lip tie is an “inconsistently defined condition” and that it is over-diagnosed in some communities [52]. An unclear relationship to breastfeeding difficulties also exists, and there is no agreement that frenectomies in primary dentition are effective in the correction of midline diastema. The current consensus of the experts on the lack of clear evidence on the correct diagnosis and intervention in the case of MLF, as well as the reported increase of referrals for MLF frenectomies during recent years, has led to the conduction of this review [53,54].

It is important to emphasize the fact that the MLF is a morphologically dynamic and diverse structure and should be evaluated accordingly. The knowledge of the elements of normal morphology and its variations at every developmental stage is essential in the correct differential diagnosis of abnormal MLF cases that require intervention. Based on this review, in infants, the normal level of attachment of the MLF is at the lower level of the alveolar process. Therefore, the structure’s clinical appearance should not be a criterion for assessing the need for surgical intervention. Future studies focusing on functional assessments of upper lip flexibility and ability to flange may provide more information and assist in the development of clinical protocols with specific guidelines.

In pre-adolescent children, the prevalence of abnormal MLF is reportedly higher than in adolescents and adults. The main reason is that, as age advances, there is a vertical growth of the alveolar process, maxillary sinus development, and intra-alveolar eruption of the permanent maxillary incisors. This change in position during growth from primary to permanent dentition was believed to be caused by the frenum’s static position while the surrounding structures grew. Therefore, early interventions in cases where there are no periodontal or psychosocial problems associated with compromised smile esthetics cannot be justified based on the currently available literature. In addition, spontaneous closure of midline diastemata, which is often the reason for intervention in this age group, is not always the result.

According to the formal American Academy of Pediatric Dentistry policy on the management of the frenulum in pediatric patients, any surgical manipulation of the frenulum should be postponed until the permanent canines erupt and only following orthodontic closure of the space, or in conjunction with orthodontic treatment [55]. The reason is that the bilateral pressure of the canines to the anterior teeth during their eruption is sufficient to close the midline diastema and physiologically constrict and eliminate hypertrophic freni.

In cases where a frenectomy is required due to periodontal, functional and/or esthetic reasons, the laser-assisted techniques appear to have some advantages over the advanced and scar-free scalpel techniques. Scalpel-based frenectomies involve cutting and suturing, various techniques like simple excision, Z-plasty, or diamond-shaped excision to improve outcomes, more post-operative discomfort, and longer healing times. Laser-based frenectomies involve cutting and cauterizing simultaneously, typically without needing sutures, less bleeding, faster healing, and less post-operative discomfort. Nevertheless, the result is not affected by the selected surgical technique if there is no residual scar tissue. Reported laser advantages were associated with reduced intraoperative bleeding, treatment time, and patient-reported peri-operative discomfort.

The main strength of this study, in addition to the systematic and thorough search of the literature, is the presentation of the results according to the age group of the patients. This is very important for the understanding of the dynamic nature of the MLF as a structure, as well as the normal morphological phenotype during development. Clinically, this will result in a more accurate diagnosis and subsequent treatment planning. Future studies should also focus more on the differential diagnosis of normal versus abnormal frenum attachment in different age or developmental stage groups, so that more specific guidelines for assessment and treatment necessity of MLFs can be developed.

### 4.2. Limitations

The main limitation of this review was the limited access of the reviewers to specific databases that they had access to via their institute. Nevertheless, manual searches of the references of the included studies, as well as of relevant reviews that were identified through the initial article screening, were conducted in an effort to identify additional studies.

## 5. Conclusions

There is a need for more objective MLF diagnostic criteria and treatment guidelines. General dentists and pediatricians, who are the main source of referrals for frenectomy procedures, need to be updated on the latest guidelines so that unnecessary surgical interventions are avoided.

## Figures and Tables

**Figure 1 diagnostics-14-01710-f001:**
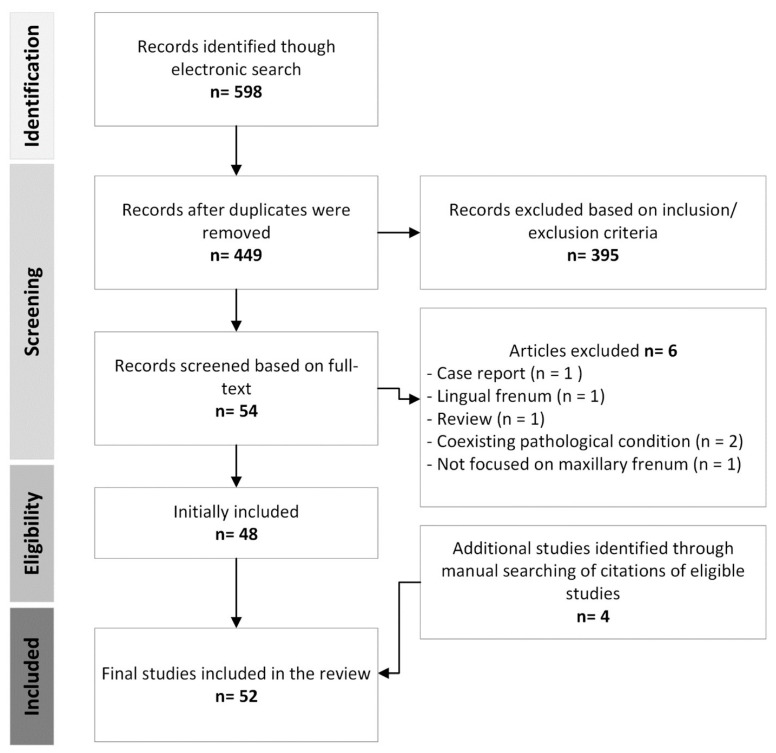
Graphic representation of article screening and eligibility assessment process.

**Table 1 diagnostics-14-01710-t001:** Eligibility criteria used for the selection of studies based on the PICOS approach.

Category	Inclusion Criteria	Exclusion Criteria
Participant characteristics	Human subjects of any gender and age	- Patients with craniofacial syndromes and/ or cleft lip palate - Animal studies
Intervention	Assessment and management of pathological maxillary frenum	- No information on the assessment and/or management of pathological maxillary frenum
Control	No inclusion criteria related to control subjects were included	No exclusion criteria related to control subjects were included
Outcome	Studies providing information on the assessment and management of pathological maxillary frenum	- Studies not providing information on the assessment and management of pathological maxillary frenum - Ongoing studies
Study design	- Randomized controlled clinical trials - Prospective controlled clinical trials - Retrospective controlled clinical trials - Case-control observational studies - Cohort studies - Cross-sectional surveys	- Systematic reviews * - Meta-analyses * - Case reports/Case series - Narrative reviews - Unsupported opinion of expert - Editor’s choices - Replies to the author/editor - Book/Conference abstracts - In vitro/In silico studies - Interviews - Commentaries

* After reviewing their lists of references for potentially eligible studies.

**Table 2 diagnostics-14-01710-t002:** Main characteristics of the included studies.

Author, Year	Title	Study Design	Country
Abullais et al., 2016 [1]	Paralleling technique for frenectomy and oral hygiene evaluation after frenectomy	Prospective Cross-sectional	India
Akpınar et al., 2016 [10]	Postoperative discomfort after Nd:YAG laser and conventional frenectomy: comparison of both genders	Prospective Cross-sectional	Australia
Baxter et al., 2023 [11]	Safety and efficacy of maxillary labial frenectomy in children: A retrospective comparative cohort study	Retrospective Cross-sectional	USA
Biradar et al., 2020 [2]	Assessment of diverse frenal morphology in primary, mixed, and permanent dentition: A prevalence study	Retrospective Cross-sectional	India
Boutsi et al., 2011 [6]	Maxillary labial frenum attachment in children	Prospective Cross-sectional	Greece
Butchibabu et al., 2014 [12]	Evaluation of patient perceptions after labial frenectomy procedure: A comparison of diode laser and scalpel techniques	Prospective Cross-sectional	India
Calisir and Ege, 2018 [13]	Evaluation of patient perceptions after frenectomy operations: A comparison of neodymium-Doped Yttrium Aluminum Garnet laser and conventional techniques in the same patients	Prospective Randomized Clinical Trial	Turkey
Cankaya et al., 2020 [14]	Evaluation of the effect of the application of hyaluronic acid following laser-assisted frenectomy: an examiner-blind, randomized, controlled clinical study	Prospective Randomized Controlled Clinical Trial	Turkey
Cinthura and Jeevanandan, 2020 [15]	Maxillary labial frenum morphology and midline diastema among children aged 3–12 years–A cross-sectional study	Prospective Cross-sectional	India
Dasgupta et al., 2017 [16]	Morphological variations of median maxillary labial frenum: A clinical study	Prospective Cross-sectional	India
Deepa, 2016 [17]	Attachment of maxillary frenum and occurrence of midline diastema in children	Prospective Cross-sectional	India
Diaz-Pizan et al., 2006 [18]	Midline diastema and frenum morphology in the primary dentition	Prospective Cross-sectional	Peru
Diercks et al., 2020 [19]	Factors associated with frenotomy after a multidisciplinary assessment of infants with breastfeeding difficulties	Prospective Case-control	USA
Haischer-Rollo et al., 2022 [20]	Superior labial frenulum attachment site and correlation with breastfeeding outcomes	Prospective Case-control	USA
Hasan et al., 2020 [21]	Pattern of distribution and etiologies of midline diastema among Kurdistan region population	Prospective Cross-sectional	Iraq
Haytac et al., 2006 [22]	Evaluation of patient perceptions after frenectomy operations: A comparison of carbon dioxide laser and scalpel techniques	Prospective Cross-sectional	Turkey
Jindal et al., 2016 [23]	Variations in the frenal morphology in the diverse population: A clinical study	Prospective Cross-sectional	India
Jonathan et al., 2018 [24]	Maxillary labial frenum morphology and midline diastema among 3 to 12-year-old school going children in Sri Ganganagar city: A cross-sectional study	Prospective Cross-sectional	India
Júnior et al., 2013 [25]	Labial frenectomy with Nd:YAG laser and conventional surgery: a comparative study	Prospective Cross-sectional	Brazil
Komori et al., 2017 [26]	Clinical study of laser treatment for frenectomy of pediatric patients	Retrospective Cohort	Japan
Kotian and Jeenenandan, 2020 [27]	Maxillary labial frenum morphology in children in Chennai population: A cross-sectional study	Prospective Cross-sectional	India
Kramer et al., 2022 [4]	Maxillary labial frenum in preschool children: variations, anomalies and associated factors	Prospective Cross-sectional	Brazil
Medeiros et al., 2015 [25]	Labial frenectomy with Nd:YAG laser and conventional surgery: a comparative study	Prospective Case-control	Brazil
Marra and Itro, 2020 [28]	Surgical management of frenula: Laser therapy compared with Z-frenuloplasty technique	Prospective Cross-sectional	Brazil
Miller, 1985 [29]	The frenectomy combined with a laterally positioned pedicle graft	Prospective Case-control	USA
Olivi et al., 2010 [30]	Er,Cr:YSGG laser labial frenectomy: A clinical retrospective evaluation of 156 consecutive cases	Retrospective Case-control	Italy
Olivi et al., 2018 [31]	Laser labial frenectomy: a simplified and predictable technique. Retrospective clinical study	Retrospective Case-control	Italy
Ozener et al., 2020 [32]	Clinical efficacy of conventional and diode laser-assisted frenectomy in patients with different abnormal frenulum insertions: A retrospective study	Retrospective Case-control	Turkey
Pandiyan et al., 2018 [5]	Clinical assessment of frenum morphology and attachment in Malaysian children	Retrospective Cross-sectional	Malaysia
Pareira Rafael et al., 2021 [33]	Longitudinal evaluation of diastema closure in patients submitted to labial frenectomy in different phases of the mixed dentition: A historical cohort	Retrospective Case-control	Portugal
Patal et al., 2015 [34]	Comparison of labial frenectomy procedure with conventional surgical technique and diode laser	Prospective Cross-sectional	India
Patel et al., 2019 [8]	Upper lip frenotomy for neonatal breastfeeding problems	Retrospective Cohort	USA
Pie-Sanchez et al., 2012 [35]	Comparative study of upper lip frenectomy with the CO_2_ laser versus the Er, Cr: YSGG laser	Prospective Randomized Clinical Trial	Spain
Popovich et al., 1977 [36]	Persisting maxillary diastema: Differentia diagnosis and treatment	Retrospective Cohort	Canada
Pransky et al., 2015 [9]	Breastfeeding difficulties and oral cavity anomalies: The influence of posterior ankyloglossia and upper-lip ties	Retrospective Cohort	USA
Rajani et al., 2018 [3]	Prevalence of variations in morphology and attachment of maxillary labial frenum in various skeletal patterns–A cross-sectional study	Prospective Cross-sectional	India
Ray et al., 2019 [37]	Anatomic distribution of the morphologic variation of the upper lip frenulum among healthy newborns	Prospective Cross-sectional	USA
Razdan et al., 2020 [38]	Maxillary frenulum in newborns: Association with breastfeeding	Prospective Cross-sectional	USA
Santa Maria et al., 2017 [39]	The superior labial frenulum in newborns: What is normal?	Prospective Cross-sectional	USA
Sarmadi et al., 2021 [40]	Evaluation of upper labial frenectomy: A randomized, controlled comparative study of conventional scalpel technique and Er:YAG laser technique	Prospective Randomized Controlled Single-blind Clinical Trial	Sweden
Schuepbach et al., 2023 [41]	Longitudinal changes of the insertion of the maxillary labial frenum in children and adolescents undergoing orthodontic treatment	Retrospective Cohort	Switzerland
Seker and Ozdemir, 2020 [42]	Assessment of pain perception after conventional frenectomy with application of cold atmospheric plasma	Prospective Case-control	Turkey
Sekowska et al., 2016 [43]	Diastema size and type of upper lip midline frenulum attachment	Prospective Cross-sectional	Poland
Sezgin et al., 2020 [44]	Evaluation of patient’s perceptions, healing, and reattachment after conventional and diode laser frenectomy: A three-arm randomized clinical trial	Prospective Randomized Clinical Trial	Turkey
Sfasciotti et al., 2020 [45]	Diode versus CO_2_ laser therapy in the treatment of high labial frenulum attachment: A pilot randomized, double-blinded clinical trial	Prospective Randomized Clinical Trial	Italy
Suter et al., 2014 [46]	Does the maxillary midline diastema close after frenectomy?	Retrospective Cross-sectional	Switzerland
Viet et al., 2019 [47]	Reduced need of infiltration anesthesia accompanied with other positive outcomes in Diode laser application for frenectomy in children	Prospective Case-control	Vietnam
Xie et al., 2022 [48]	Comparative frenectomy with conventional scalpel and dual-waved laser in labial frenulum	Prospective Randomized Clinical Trial	China
Yadav et al., 2019 [49]	Frenectomy with conventional scalpel and Nd:YAG laser technique: A comparative evaluation	Prospective Cross-sectional	India
Zakirulla et al., 2021 [50]	Maxillary midline frenum morphology and its variations in Saudi children	Prospective Cross-sectional	Saudi Arabia
Zen et al., 2020 [51]	Identification of oral cavity abnormalities in pre-term and full-term newborns: a cross-sectional and comparative study	Prospective Cross-sectional	Brazil

**Table 3 diagnostics-14-01710-t003:** Summary of the results of the included studies.

Author, Year	Study Objective	Sample Size (Females/Males)	Mean Age and/or Range	Pertinent Results Summary
Abullais et al., 2016 [1]	To compare the post-operative response and oral hygiene maintenance of subjects who received frenectomies with those of the paralleling and conventional technique.	20 (8:12)	20–35 years	The paralleling frenectomy technique, when compared to the conventional scalpel technique, demonstrated less patient-reported post-operative discomfort and functional complications during the healing phase.
Akpınar et al., 2016 [10]	To determine if pain was perceived differently between males and females after scalpel and laser frenectomy procedures.	89	29 years	Nd:YAG laser frenectomy technique provided better post-operative comfort, especially in females, in terms of pain, chewing, and speaking than the scalpel technique did during the first week post-operatively.
Baxter et al., 2023 [11]	To assess the safety of maxillary frenectomy before orthodontic treatment and whether the size of the diastema is affected by early treatment.	109 (66:43); Primary dentition: 95 (58:37); Mixed dentition: 14 (8:6)	0.8–11 years; Primary dentition: 1.9 ± 1.5 years; mixed dentition: 8.1 ± 1.3 years	No adverse outcomes were reported post-operatively, apart from minor pain and swelling. A decrease in diastema width was observed in 94.7% of the 109 patients in primary dentition.
Biradar et al., 2020 [2]	To provide information regarding the normal variation of the frenal morphology and attachment in primary (Group I), mixed (Group II), and permanent (Group III) dentition.	1800; Group I: 600 (273:327); Group II: 600 (281:319); Group III: 600 (277:323)	3–17 years	There was a highly statistically significant difference in the type of frenal morphology and frenal attachment in all groups. The prevalence of simple frenum increases from primary dentition to permanent dentition, whereas PTF decreases as age increases. This study reveals a high prevalence of gingival attachment followed by papillary attachment. There was no significant association of frenal type with gender.
Boutsi et al., 2011 [6]	To examine the prevalence of the various types of maxillary labial frenum attachment among children of different ethnic backgrounds.	226 (106:119)	8.5 ± 3.0 years	Ethnic background and gender are not associated with frenum type. Age was strongly correlated.
Butchibabu et al., 2014 [12]	To compare post-operative pain with YAG laser and scalpel frenectomy.	10 (4:6)	18–30 years	The pain and discomfort compared for the two groups were different for all the days measured after surgery.
Calisir and Ege, 2018 [13]	To assess the post-operative pain levels after scalpel and YAG laser frenectomy.	40 (20:20)	22–23 years; females: 22.1 ± 3.57 years, males: 23.1 ± 3.44) years	Patients who received YAG laser frenectomy had lower levels of pain and most patients preferred the laser frenectomy.
Cankaya et al., 2020 [14]	To evaluate whether hyaluronic acid can aid post-operative healing.	40 (14:26); Hyaluronic acid: 20, Control: 20	18–40 years	Hyaluronic acid was found to decrease wound irritation and increase patient satisfaction after frenectomy.
Cinthura et al., 2020 [15]	1. To estimate the prevalence of different morphologic types of maxillary labial frenum among children ages 3–12 years. 2. To assess the association between frenum morphology and midline diastema in children. 3. To assess the association between age and midline diastema in children	100 (48:52)	3–12 years	An abnormal MLF can be related to persistent midline diastemata. Orthodontic closure of the diastema is usually deferred until the eruption of the permanent canines but can start earlier in cases with very large diastemata.
Dasgupta et al., 2017 [16]	To determine the prevalence of morphologic variations and classify MLF. The study also compares the morphological variations of MLF among different ages and genders.	1400 (700:700)	3–12 years	The most common type of frenum was the simple type. A statistically significant difference was found in the proportions of different types of frenum among the different age groups and the different sites of the presence of frenal attachments. No statistically significant differences were found between male and female subjects.
Deepa, 2016 [17]	To identify the attachment of the maxillary frenum and the presence of midline diastema in school children.	200 (80:120)	7–15 years	The most common type of frenal attachment diagnosed in this study was the mucosal type. According to the Placek classification, frenum attachment types 2 and 3 are associated with the development of midline diastema.
Diaz-Pizan et al., 2006 [18]	To determine the prevalence of different types and insertions of labial and midline diastema in Peruvian children 0–6 years of age.	1355 (658:697)	0–6 months to 6 years	The most common frenum type was “simple frenum”. An inverse correlation between gingival insertion and midline diastema was detected.
Diercks et al., 2020 [19]	To determine whether the need for frenectomy procedures can be evaluated with the use of comprehensive feeding evaluations.	11	47 days	Functional assessment, rather than just upper lip frenulum appearance, is important for identifying patients who may also benefit from a lip frenotomy to help with feeding difficulties.
Haischer-Rollo et al., 2022 [20]	To investigate normal MLF attachment sites and grades with the use of a modification of an existing classification system and evaluate the association with the outcomes of breastfeeding.	208 (117:91)	39.1 weeks	This study shows no correlation between MLF attachment grade and breastfeeding outcomes, and the results do not support the frenectomy decision-making based on this criterion.
Hasan et al., 2020 [21]	To assess the prevalence and etiological factors of midline diastema in the population of the Kurdistan Region of Iraq among different age groups and sex.	1021 (483:538)	19.6 ± 4.8(13–35) years	The main causes of midline diastema in females were thumb sucking and missing lateral incisors, and high labial frenum and supernumerary teeth in males.
Haytac et al., 2006 [22]	To measure the degree of pain after frenectomy with a CO_2_ laser or a scalpel technique.	40 (24:16)	18–26 years	Patients who received a frenectomy performed with the CO_2_ laser had less post-operative pain and required fewer analgesics compared to those who had the scalpel technique.
Jindal et al., 2016 [23]	To investigate the prevalence of frenulum variations in a diverse ethnic population.	500 (285:215)	16–40 years	MLF has diverse morphology. Normal frenum was most common, followed by a frenum with a nodule, while a frenum with an appendix was found to be the least common. The prevalence of midline diastema was found to be higher in the papillary penetrating type as compared to papillary and gingival attachment.
Jonathan et al., 2018 [24]	1. To estimate the prevalence of different morphologic types of maxillary labial frenum among children aged 3–12 years. 2. To determine the relationship between the level of insertion of the frenum and the age of the child. 3. To evaluate the correlation between frenum morphology, insertion, and midline diastema	1200	25.2% were 3–5 years old; 40.1% were 6–9 years old; 34.5% were 10–12 years old	The presence of an abnormal MLF attachment can cause midline diastemas. Tooth movement usually waits until the eruption of permanent canines.
Júnior, R.M. et al., 2013 [25]	This study aimed to compare pre-, trans-, and post-surgical parameters of laser and scalpel frenectomies.	40	20.9 (8–51) years	The YAG laser technique had less post-operative bleeding, but there was no difference in pain, and oral function was different between the two methods.
Komori et al., 2017 [26]	To suggest the correct timing of laser frenectomies in a pediatric population.	8	4–9 years	Frenectomies were performed with the use of a CO_2_ laser. No intraoperative adverse events were reported, and the procedures were quickly and safely performed. Re-adhesion was noted in one patient.
Kotian and Jeenenandan, 2020 [27]	To determine the prevalence of types and attachment levels of the maxillary labial frenulum in a Chennai population.	200 (90:110)	3–10 years	Frenum morphology varies in children. Mucosal and gingival attachment is more prevalent in children.
Kramer et al., 2022 [4]	To investigate the morphological and attachment variations of the MLF in preschool children.	1313	0–5 years	The most prevalent patterns were simple MLF and gingival attachment. Morphological abnormalities were found in 21.6% of the preschoolers and 25.4% exhibited abnormal frenal attachment. Abnormalities in MLF morphology were more prevalent among girls and a significant reduction was found with the increase in age. Attachment abnormalities were significantly more prevalent among girls, the white ethnic group, and children who used a pacifier and reduced significantly with age.
Madeiros et al., 2015 [25]	To compare clinical parameters before, during, and after labial frenectomies performed with conventional surgery and with a Nd:YAG laser.	40; 18 (24:16); Surgery: 22 (11:11); Laser: 18 (13:5)	10.3 (8–51)	The use of Nd: YAG laser for oral frenectomies presents with the following advantages: absence of trans-operative bleeding, no requirement for sutures and a significant reduction in surgical time in comparison with conventional surgery. No superiority with regard to post-operative pain and oral function was observed.
Marra and Itro, 2020 [28]	To compare intra- and post-operative consequences after a z-frenuloplasty and laser for maxillary labial and lingual frenectomies.	120	11 years & 2 months (9 to 14 years)	Z-frenuloplasty and laser treatment are both valid for frenectomies. Laser offers more advantages: lower need for anesthesia, hemostasis, no suturing, faster healing, and fewer limitations in speech and nutrition.
Miller, 1985 [29]	To assess a frenectomy technique that combines the preservation of the dental papilla and a laterally positioned pedicle graft from a functional and esthetic perspective.	24	N/A	The results with the use of the presented frenectomy technique were satisfactory with only 3 out of 24 cases presenting with minimal relapse, whereas the esthetic results were superior to that obtained with the classic frenectomy technique.
Olivi et al., 2010 [30]	To present the retrospective assessment of a group of pediatric patients following frenectomies performed with an Er,Cr:YSGG laser.	43 (70–73)	7–11 years	The Er,Cr:YSGG laser led to a considerable reduction in operating time, eliminated or reduced the amount of local anesthetic used during the procedure, significantly reduced the need for surgical sutures. The pediatric patients tolerated the post-operative phase, eliminating the need for analgesics. Furthermore, there was a minimal scar tissue, and low recurrence rate.
Olivi et l., 2018 [31]	To propose a surgical frenum repositioning technique and identify clinical indications for MLF frenectomy associated with early orthodontic therapy, to justify early frenum repositioning in children.	20	8–10 years	Post-operatively, all patients reported no pain and no bleeding. No relapse occurred 4 years after frenectomy. The use of Er:YAG laser significantly reduced the operating time, and the amount of local anesthetic and eliminated the need for surgical sutures.
Ozener et al., 2020 [32]	To compare the recurrence of frenulum attachment in patients with different abnormal frenulum insertions after scalpel and diode laser-assisted frenectomies and evaluate clinical periodontal parameters and the presence of diastema.	70 (23:47); Surgery: 36 (15:21), Laser: 34 (8:26)	35.24 ± 11.69; Surgery: 35.53 ± 10.89 (18–64); Laser: 34.94 ± 12.66 (18–60)	32.9% presented gingival, 38.6% papillary, and 28.6% papilla-penetrating frenulum attachments. PI and GI were significantly higher in the conventional group, whereas PD was similar at 6 weeks.
Pandiyan et al., 2018 [5]	To determine the prevalence of various types and attachment levels of maxillary frenum in Malaysian children.	200 (100:100)	8.6 ± 3.2 years	Frenum attachment changes in children with age. Gender plays no role in portraying difference. Clinicians must be able to identify different types of attachments to avoid surgical intervention during this time of development.
Pereira Rafael et al., 2021 [33]	To evaluate the spontaneous closure of the interincisal diastema in patients submitted to upper labial frenectomy during the mixed dentition.	53	6–12 years	There is no correlation between the time of surgery intervention and diastema closure. It was observed that 44% showed total diastema reduction and 52% presented a reduction of the diastema when the frenectomy was performed after lateral eruption. Intervention during the mixed dentition led to a partial diastema reduction in 80% of the cases. Intervention in mixed dentition resulted in 12% of the patients without diastema reduction, while 36% presented total reduction. It was not possible to observe any association between the exposure factors and the outcomes.
Patal et al., 2015 [34]	To compare the level of pain that the patient perceived during a frenectomy with either the YAG laser or scalpel.	20	16–40 years	Diode lasers provide a better outcome for the patient regarding discomfort and healing.
Patel et al., 2019 [8]	To raise awareness of upper lip ties as a potential contributor to neonatal breastfeeding problems and share a technique for in-office frenotomy.	34	39 (36–42) weeks	An improved latch was noted by 82% of mothers and 54% felt their infants received more milk. Local pain was reported for 82% of children, either mild (64%) or moderate (18%) in severity. When present, the lip pain resolved within 24 h for 72% of children, with the remainder resolving in a few days or within 1 week. Recurrence of lip tie was reported by 9% of mothers; no infection or other complications were reported. There was no significant relationship between the change in reported latch and child age, gender, or the presence of supplemental bottle feeding.
Pie-Sanchez et al., 2012 [35]	To compare upper lip frenulum reinsertion, bleeding, surgical time, and surgical wound healing in frenectomies performed with the CO_2_ laser versus the Er, Cr:YSGG laser	50 (22:28)	11.3 ± 0.8 years	Insertion of the frenulum migrated to the mucogingival junction as a result of using both laser systems in all patients. Only two patients required a single dose of paracetamol, one of either study group. CO_2_ laser registered improved intraoperative bleeding control results and shorter surgical times. On the other hand, the Er,Cr:YSGG laser achieved faster healing.
Popovich et al., 1977 [36]	To provide clinical guidelines for the diagnosis of cases with persistent midline diastema.	471	9 and 16 years	Low thick frenum attachment occurs much more frequently with persistent diastema. The frenum can actively maintain the diastema but only in association with other factors. Generalized spacing appears to be the primary cause and the related low, thick frenum and suture type make only a minor contribution. Frenectomy is only necessary in very few cases and even then, it should be done after the central incisors have been approximated and held together, otherwise, scar tissue may develop and continue to hold the incisors apart.
Pransky et al., 2015 [9]	The objective of this study was to describe the experience of the team at a high-volume breastfeeding difficulty clinic with a focus on posterior ankyloglossia and upper-lip ties	14	Infants	Upper lip-tie release led to improved breastfeeding in all cases. However, causation cannot be implied from the current study.
Rajani et al., 2018 [3]	To assess the significance of various types of frenum based on the attachment site and morphology in different skeletal patterns.	150 (Class I = 50, Class II = 50, Class III = 50)	13–30 years	Abnormal frenum categories based on its location and morphology were significantly more prevalent in Class III and different from the other groups. Papillary and papillary penetrating types are significantly associated with skeletal class III pattern and midline diastema.
Ray et al., 2019 [37]	To measure the variations in length, thickness, and attachments of the MLF in healthy newborns and to identify which anatomic measurements could be used in further research investigating the MLF.	150 (77:73)	Newborns	Variations in MLF morphology were identified, and some combination of the stated measurements may be used to create a more robust classification.
Razdan et al., 2020 [38]	To relate maxillary and lingual frenulum configuration to breastfeeding success.	161 (79:82)	Newborns	MLF score did not significantly correlate with LATCH score. Mothers experienced with breastfeeding had better LATCH score.
Santa Maria et al., 2017 [39]	To develop a classification system for superior labial frenula and to estimate the incidence of different degrees of attachment.	100 (44:56)	Newborns	Interrater agreement scores were low when it came to the assessment of the MLF using the Kotlow classification system. The agreement rates improved when the classification system was simplified.
Sarmadi et al., 2021 [40]	To compare frenectomies performed using Er:YAG laser with those using the scalpel technique. Comparisons were of patients’ experiences, treatment times, bleeding during treatment and wound healing.	39; Surgery: 19 (18:1); Laser: 20 (16:4)	7–19 years; Surgery: 11.5 (8–13) years, Laser: 9 (8–11) years	Significantly reduced surgical time and bleeding were observed with the scalpel surgery. Directly after surgery, the wound area was significantly larger in the laser group, but at the 5-day evaluation, no difference could be observed between the groups. Patients were satisfied with both methods, giving them the same assessments.
Schuepbach et al., 2023 [41]	To evaluate the potential vertical changes in the insertion of the MLF in growing children and to associate these changes with the vertical growth of the dentoalveolar process and thelower third of the face.	33 (17:16)	Females: 11.2 years; Males: 12.4 years	The MLF remains stable in reference to the palatal plane, whereas the maxillary incisal edge moves away from it. The distance from the MLF to the incisal edge increases over the childhood years. The dentoalveolar process undergoes vertical growth without affecting the insertion of the frenum.
Seker and Ozdemir, 2020 [42]	To look at how Cold Atmospheric Plasma (CAP) is applied after traditional frenectomy.	87 (49:38)	18–36 years	Frenectomy surgery with CAP application had positive results on pain reduction and wound healing.
Sekowska et al., 2016 [43]	To assess frenum attachment in patients with diastema and investigate if the type of upper lip frenum attachment affects the diastema width.	102	MMD group: 23.2 ± 7; Controls: 21.6 ± 6	Patients with diastema often have oversized MLFs.
Sezgin et al., 2020 [44]	To compare the scalpel and diode laser techniques in terms of patients’ perceptions, epithelization, reattachment, and periodontal clinical parameters in the treatment of abnormal papillary frenum.	48 (27:21); Surgical: 16, Laser: 16, Laser and incision = 16	33.15 ± 10.02 (18–54) years	The use of a diode laser in frenectomy provides less post-operative pain and discomfort during speaking and chewing in patients with abnormal papillary frenum attachments. Both techniques led to the same epithelization period and prevented the frenum reattachment.
Sfasciotti et al., 2020 [45]	To compare the labial frenectomy with the use of Diode and CO2 laser techniques in pediatric patients with a high labial frenulum attachment.	26	9 (7–12) years	Laser-based techniques had better post-operative results.
Suter et al., 2014 [46]	To evaluate the maxillary frenum opening in patients with and without orthodontic treatment.	59 (42:17)	13.2 (7.8–39) years	Closure of the maxillary midline diastema with an abnormal frenum is more predictable with frenectomy and orthodontic treatment than with frenectomy alone.
Viet et al., 2019 [47]	To evaluate the need for local anesthesia, intraoperative bleeding control, and post-operative pain and wound healing in children during laser-assisted MLF frenectomies.	30 (10:20)	10.5 ± 2.3 years	Diode Laser in MLF frenectomy for children reduces the requirement of local anesthesia and increases the positive behavior of children during the procedures. The post-operative healing is mostly uneventful.
Xie et al., 2022 [48]	To compare MLF frenectomy laser (Er:YAG and Nd:YAG) and scalpel techniques.	34	5–10 years	The Nd:YAG laser group had better chewing and speaking scores than the scalpel group.
Yadav et al., 2019 [49]	To compare the scalpel versus the Nd:YAG MLF frenectomies, based on post-operative pain levels, intraoperative bleeding, healing outcome, and need for analgesics.	20	N/A	Nd:YAG lasers can be considered a viable alternative to the scalpel technique for MLF frenectomy. Lasers have the advantage of better patient acceptance due to reduced pain perception, post-operative discomfort, and reduced intraoperative bleeding.
Zakirulla et al., 2021 [50]	To evaluate and determine the different types and attachment levels of MLF in Saudi children.	200 (106:94)	7.7 ± 3.2 (2–14) years	Gingival frenum was the most detected followed by papillary frenum, papillary penetrating, and mucosal. 67.5% of children aged 11–14 years had gingival frenum compared to 67.2% of those who aged 2–5 years and 58.8% of children aged 6–10 years. Females had more common gingival frenum morphology compared to males.
Zen et al., 2020 [51]	To compare maxillary labial frenum topography between pre-term (PT) and full-term newborns (FT).	PT: 74 (32:42); FT: 100 (49:51)	PT: 24–36 weeks; FT: 37–42 weeks	Maxillary labial frenum insertion in the piriform papilla was more prevalent in PT newborns.

## Data Availability

No new data were created as part of this study.

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
