# Peer review of "Assessment and Management of Maxillary Labial Frenum—A Scoping Review"

_diagnostics, 2024, doi:10.3390/diagnostics14161710_

Round 1

Reviewer 1 Report

Comments and Suggestions for Authors

This review systematically reviews the research progress of the maxillary labial frenum, which is fair, comprehensive, and reasonable, and has certain guiding significance for clinical practice. It is recommended to publish it.

Author Response

Comment 1: This review systematically reviews the research progress of the maxillary labial frenum, which is fair, comprehensive, and reasonable, and has certain guiding significance for clinical practice. It is recommended to publish it.

Response 1: We would like to thank the reviewer for his positive feedback.

Reviewer 2 Report

Comments and Suggestions for Authors

The authors reviewed and meta-analysis articles associated with assesment and management of maxillary labial frenum. This review provide systemic concepts with recent assessment and management for dentist. The results and conclusions may help dentist a evidence for dealing and facing maxillary labial frenum.  I have one personal question. Why did the authors use the protocol developed  by the Joanna Briggs Institute and this protocol has not been registed online? Is this protocol still under development or this protocol is not already well accepted by researchers?

The English editing is well. There is less than 9% plagarism using editing software checked. If the authors can explained why they use the protocol well, the article may be considered ready for publish. 

Author Response

Comment 1: Why did the authors use the protocol developed by the Joanna Briggs Institute...?

Response 1: We would like to thank the reviewer for this comment and the opportunity to clarify our methodology. This paper is a scoping review. The Joanna Briggs Institute (JBI) was the first to publish a formal guidance document for the conduct of scoping reviews. The JBI Scoping Review Methodology Group provides comprehensive resources, including templates, guidance of data extraction, analysis and presentation. The PRISMA reporting guidelines for systematic reviews were updated based on the same protocol to create the PRISMA-ScR (PRISMA extension for Scoping Reviews), which were used for the conduction and reporting of this review. These are the reasons why we decided to follow the specific protocol. 

Comment 2: ...and this protocol has not been registered online?

Response 2: To our knowledge, there is no formal protocol registration platform for scoping reviews yet.

Comment 3: Is this protocol still under development or this protocol is not already well accepted by researchers?

Response 3: We followed the protocol created by the JBI, which is a validated protocol. Please see the original methodology open access publication: Peters, Micah D.J. BHSc, MA(Q), PhD1; Godfrey, Christina M. RN PhD2; Khalil, Hanan BPharm, MPharm, PhD3; McInerney, Patricia PhD4; Parker, Deborah5; Soares, Cassia Baldini RN, MPH, PhD6. Guidance for conducting systematic scoping reviews. International Journal of Evidence-Based Healthcare 13(3):p 141-146, September 2015. | DOI: 10.1097/XEB.0000000000000050. It is a widely accepted protocol by researchers worldwide. 

Reviewer 3 Report

Comments and Suggestions for Authors

Dear Authors,

Congratulations on the job you have done and presented in this manuscript. I believe that your work might be of high interest for the general reader, however the are some concerns that require revision before consideration for publication in a such high quality journal. Please see the attachment.

Author Response

Please find the responses to the comments in the respective comment boxes initially added by the reviewer and numbered as instructed. 

Reviewer 4 Report

Comments and Suggestions for Authors

Dear Authors, 

Thanks for the opportunity to review your paper titled " Assessment and management of maxillary labial frenum" and I appreciate the effort you've put into your research. 

The topic is interesting and the main aim of this study is  to summarize the current evidence on the assessment and management of abnormal MLF.

The research falls within the journal's scope. However, I would like to suggest some major revisions that I believe could enhance the clarity of your paper:

1. the introduction should be improved according to the recent literature on this topic

2. although the methodology is correct, the scientific premises should be included and the reasoner choosing to conduct a scoping review instead of other type of review (narrative, systematic) should be discussed 

3. in the methods please report the PICO question clearly: what is the aim ? please report it in evidence

4. the results are confusingly reported: Please summarize the characteristics of the included studies in a single table where the title is not necessary (it is in the references list). You can use the following columns: author and year, country, type of study design, patients characteristics, outcomes and main results. 

5. please improve the discussion according to other publications and report strength and limitations of your article 

Author Response

Comment 1: The introduction should be improved according to the recent literature on this topic

Response 1: We would like to thank the reviewer for this suggestion. However, since this is a review study, dedicated in the presentation of the past and most recent literature in this topic. Therefore, we would rather present this information in the results section instead. Based on the methodological guidelines followed in this study, the rationale and objective of the review are the two items required for reporting. In addition, according to our knowledge and after thoroughly searching the literature prior and during the conduction of this study, there are no similar scoping or systematic reviews published on this specific topic either, whose results we could summarize in this section. Therefore, we have to kindly decline this suggestion by the reviewer at this stage. 

Comment 2: although the methodology is correct, the scientific premises should be included and the reasoner choosing to conduct a scoping review instead of other type of review (narrative, systematic) should be discussed 

Response 2: We would like to thank the reviewer for this suggestion. A clarifying methodological statement has been added to the main manuscript to address this comment: "The specific type of review was selected because it allows for a systematic review of the literature with the aim to provide an overview of the available research evidence and elucidate potential gaps in the existing knowledge."

Comment 3: in the methods please report the PICO question clearly: what is the aim ? please report it in evidence

Response 3: We thank the reviewer for this suggestion. The PICO question is more appropriate for clinical questions, often addressing the effect of an intervention/therapy/treatment. In systematic review studies, the PICOS approach is often implemented for the reporting of the eligibility criteria. This is the approach the was followed in this study as well. We have revised Table 1 accordingly to clarify the use of this standardized approach. We hope that this change is satisfactory for the reviewer. The aim of the study is stated in the last sentence of the introduction. 

Comment 4: the results are confusingly reported: Please summarize the characteristics of the included studies in a single table where the title is not necessary (it is in the references list). You can use the following columns: author and year, country, type of study design, patients characteristics, outcomes and main results.

Response 4: We would like to thank the reviewer for this suggestion. We had initially attempted to create a single table with the suggested content. However, the table had too many columns (six to be specific, even with the exclusion of the title column) and was impossible to fit in the manuscript in the format requested by the journal (portrait orientation) and we had to split it in two separate tables. We hope that the reviewer understands this practical issue and will accept the current tables as they are. 

Comment 5: please improve the discussion according to other publications and report strength and limitations of your article 

Response 5: We would like to thank the reviewer for this suggestion. As we mentioned in the first comment, according to our knowledge there are no similar scoping or systematic reviews published on this specific topic, whose results could be summarized in this section. However, we are open to other specific suggestions that the reviewer might have for the improvement of the discussion. We have also added a paragraph discussing the strengths of this study in the end of the discussion in the revised manuscript: "

The main strength of this study, in addition to the systematic and thorough search of the literature, is the presentation of the results according to the age group of the patients. This is very important for the understanding of the dynamic nature of the MLF as a structure, as well as the normal morphological phenotype during development. Clinically, this will result in a more accurate diagnosis and subsequent treatment planning. Future studies should also focus more on the differential diagnosis of normal versus abnormal frenum attachment in different age or developmental stage groups, so that more specific guidelines for assessment and treatment necessity of MLFs can be developed."

The limitations are discussed in a separate section after the discussion. 

Round 2

Reviewer 3 Report

Comments and Suggestions for Authors

Dear Authors, 

The revised version is improved and I believe that your paper can be of high interest for the general reader, therefore I will recommend publication of your job. Congratulations!

Reviewer 4 Report

Comments and Suggestions for Authors

The authors followed all my suggestions